# The Impact of Microglia on Neurodevelopment and Brain Function in Autism

**DOI:** 10.3390/biomedicines12010210

**Published:** 2024-01-17

**Authors:** Yuyi Luo, Zhengbo Wang

**Affiliations:** 1State Key Laboratory of Primate Biomedical Research, Institute of Primate Translational Medicine, Kunming University of Science and Technology, Kunming 650500, China; luoyuyi2021@163.com; 2Yunnan Key Laboratory of Primate Biomedical Research, Kunming 650500, China

**Keywords:** microglia, neurodevelopment, neurogenesis, synaptic pruning, pathological mechanism

## Abstract

Microglia, as one of the main types of glial cells in the central nervous system (CNS), are widely distributed throughout the brain and spinal cord. The normal number and function of microglia are very important for maintaining homeostasis in the CNS. In recent years, scientists have paid widespread attention to the role of microglia in the CNS. Autism spectrum disorder (ASD) is a highly heterogeneous neurodevelopmental disorder, and patients with ASD have severe deficits in behavior, social skills, and communication. Most previous studies on ASD have focused on neuronal pathological changes, such as increased cell proliferation, accelerated neuronal differentiation, impaired synaptic development, and reduced neuronal spontaneous and synchronous activity. Currently, more and more research has found that microglia, as immune cells, can promote neurogenesis and synaptic pruning to maintain CNS homeostasis. They can usually reduce unnecessary synaptic connections early in life. Some researchers have proposed that many pathological phenotypes of ASD may be caused by microglial abnormalities. Based on this, we summarize recent research on microglia in ASD, focusing on the function of microglia and neurodevelopmental abnormalities. We aim to clarify the essential factors influenced by microglia in ASD and explore the possibility of microglia-related pathways as potential research targets for ASD.

## 1. Introduction

Autism spectrum disorder (ASD) is a complex and heterogeneous developmental disorder. Approximately 1 in 44 children are diagnosed [1]. The affected individuals mainly present with early social dysfunction and deficits in repetitive behaviors and interests [2]. Current research suggests that ASD can be classified into two types: syndromic and nonsyndromic autism. Syndromic autism usually refers to mutations in specific genes or genomes, resulting in neurological syndromes such as fragile X syndrome, tuberous sclerosis, Rett syndrome (RTT), Phelan–McDermid syndrome, and Angelman syndrome. Nonsyndromic autism comprises the majority of autism cases and is not associated with other neurological diseases but is linked to a number of genes (such as CDH8, ADNP, etc.) [3]. ASD is a multifactorial neurodevelopmental disorder influenced by genetic and environmental factors. Studies have indicated that 10–20% of individuals with ASD have genetic factors, including gene defects and chromosomal abnormalities [4]. The results of large-scale genetic studies indicate that there are hundreds of risk genes associated with ASD, which are believed to contribute to brain development, neuronal synaptic function, protein translation, and the Wnt signaling pathway. These pathways have been identified as the three major cellular pathways affected by mutations in ASD-related genes. Such alterations impact the normal function of neurons and neural circuits, playing a significant role in the progression and pathogenesis of ASD [5]. This molecular pathology involves the upregulation of microglia, astrocytes, and neuroimmune genes, the downregulation of synaptic genes, and reduced gene expression gradients in the cortex. These changes in gene expression primarily affect excitatory neurons and glial cells [6].

Microglia are a type of mononuclear macrophages that reside in the central nervous system (CNS). They are widely distributed throughout the CNS and account for 5–15% of the total number of cells in the brain [7]. Single-cell data have shown differences in gene expression patterns between fetal and adult microglia, indicating that microglia play an important role in CNS development and function at specific time points [8,9]. As resident immune cells in the CNS, microglia can polarize into different proinflammatory (M1) or anti-inflammatory (M2) phenotypes in order to maintain the integrity of the blood–brain barrier (BBB), with M1 proinflammatory microglia leading to BBB dysfunction and vascular “leakage”, and M2 anti-inflammatory microglia acting as protectors at the BBB. In addition, they readily monitor pathogens in the CNS, promote and repair neuroinflammation, and participate in immune-related signaling pathways [10,11]. Under physiological conditions, microglia constantly monitor the CNS microenvironment and quickly respond to any situation that occurs in the brain [12]. When external stimuli or pathogens affect the CNS, microglia transform into active phagocytic microglia [13] and migrate and accumulate rapidly at the damaged area [14]. Currently, more and more studies have found that damage to microglia plays a key role in the development of ASD. Some reports suggest that many pathological phenotypes of ASD may be caused by abnormalities in immune cells, specifically microglia. These brain immune cells typically prune unwanted synapses or brain connections during early development. Mouse models have determined that damage to microglia affects its function of synaptic pruning, leading to deficits in social behavior. These findings suggest that abnormal microglial function can lead to ASD-like behaviors. The mechanisms related to microglial abnormalities in ASD may become a promising direction for preventing ASD and developing ASD drugs [15]. In this article, we reviewed current research, focusing on the role of microglia in neurogenesis and development, and its involvement in the pathogenesis of autism spectrum disorders. We comprehensively elaborated on the abnormal changes in microglia in autism spectrum disorders, including their own functions, their effects on other cells in the nervous system, and their interactions. We also discussed the impact of microglia on the pathological process of autism spectrum disorders, aiming to provide a foundation and deeper insights into the role of microglia in autism spectrum disorders.

## 2. The Source of Microglia

Initially, most scientists believed that microglia, like other cell types in the brain, were derived from the ectoderm [16]. However, later studies found that the loss of PU.1, a key transcription factor in bone marrow cells, led to the loss of microglia in the mouse brain [17]. Microglia also share many characteristics with macrophages derived from bone marrow [18]. It was further demonstrated that microglia may originate from bone marrow macrophages that penetrate the brain meninges during late-stage embryonic development [19]. As one of the main cell types in the CNS, microglia are widely distributed throughout all areas of the brain, and the number and density of microglia vary in different brain regions [20]. Microglia have two main roles in the CNS: (1) as resident tissue macrophages in the brain, they play an immune defense function [21]; (2) microglia interact with the microenvironment in the brain to regulate neurogenesis and synapse formation, maintain the survival of neurons, and participate in maintaining CNS homeostasis [22]. Given the important role of microglia in the CNS, their involvement has been found in the pathological processes of many neurological diseases [23].

## 3. Neurodevelopmental Abnormalities on ASD Caused by Microglia

In the early stages of CNS development, neural progenitor cells (NPCs) first differentiate into neurons, while microglia migrate into the nervous system. At this time, oligodendrocytes and astrocytes have not yet formed. Therefore, the role of microglia as the main glial cell type in the CNS during CNS development cannot be ignored [24]. Studies have shown that microglia play a critical role in controlling neurogenesis and ultimately regulating the size of the cerebral cortex and other CNS structures by directly regulating neurogenesis through phagocytizing neural precursor cells [5]. During development, microglia play a role in promoting neurogenesis and synaptic pruning [25]. This suggests that microglia also participate in the neurodevelopmental processes of ASD and may play a key role. In this article, we briefly discuss the factors related to microglia and their involvement in neurogenesis and developmental abnormalities in ASD.

### 3.1. Microglia Affect Neurogenesis in ASD

Studies have shown that microglia play a critical role in controlling neurogenesis and ultimately regulating the size of the cerebral cortex and other CNS structures by directly regulating neurogenesis through phagocytizing neural stem cells (NSCs) [26]. A large number of neural lineage cells, including NPCs, undergo programmed death as a quality control mechanism, and only the last surviving NPCs can continue to differentiate into mature neurons [27]. Microglia exert this phagocytosis after programmed cell death in the nervous system to remove apoptotic cells and regulate the number of NSCs in the CNS, aiming to maintain neurogenesis in the nervous system [28]. Neurogenesis in immunodeficient mice is somewhat impaired even under appropriate conditions, suggesting that microglia are required for adult neurogenesis [29]. In vitro studies have found that using a microglia-conditioned medium for in vitro neuron culture can promote the proliferation of NSCs [30]. Another study has shown that coculture of wild-type microglia with neural progenitor cells isolated from the microglia-deficient cerebral cortex also ameliorated abnormalities in neural progenitor cell proliferation [31]. More and more research suggests that ASD is associated with neuroinflammation, abnormal neurotransmitter levels, and neurogenesis. Early inflammatory responses, such as maternal immune activation (MIA) and childhood infections, are among the most recognized environmental factors leading to neurodevelopmental disorders [32]. Some ASD patients’ gray matter volume of the brain is increased, indicating abnormal brain development [33]. Studies have also reported that MIA can cause ASD-like behaviors in offspring, trigger a chronic proinflammatory response, and overactivate microglia during fetal development. Activated microglia can negatively regulate synaptic pruning and neurogenesis by changing the expression of key phagocytic genes. This overactivation can lead to neuronal defects, brain structural changes, and behavioral abnormalities in offspring [34] (Figure 1). The number of microglia in fetal and neonatal mice stimulated by MIA increased dramatically, and Ki67+/Nestin+ and Tbr2+ NPCs in the subventricular zone also increased significantly. This indicates that during pregnancy, MIA triggers neuroinflammation in the early stages of neural development, which promotes an abnormal activation of microglia, leading to abnormal neurogenesis, impaired brain function, and the emergence of ASD behaviors in offspring [35]. Minocycline is a tetracycline derivative that inhibits microglial activation and has been proven to have anti-inflammatory properties and neuroprotective effects. The administration of minocycline can inhibit the activation of microglia, improve neurogenesis in the hippocampus of ASD mouse models, and reduce ASD-like behaviors. This suggests that minocycline may be a potential strategy to improve ASD symptoms [36]. The above studies illustrate that excessive activation of microglia in ASD may lead to damaged neurogenesis, neuronal defects, and abnormal brain function. The use of minocycline can improve the pathological phenotype of ASD by inhibiting microglial activation.

### 3.2. Microglia Affect the Neural Circuits on ASD

#### 3.2.1. Synapse Formation and Pruning via the Complement Pathway

In vitro studies have found that conditioned medium derived from mature microglia can promote a faster maturation of neurons cultured in vitro [37]. The normal formation and pruning of synapses are necessary steps for neurons to mature. Microglia are primarily active in areas close to synapses in the early postnatal and adult cerebral cortex under physiological conditions [38,39]. Studies have shown that synapse formation in mice is affected after microglia loss [40]. The influence of microglia on synapse formation may be achieved through contact with dendritic spines. In vivo imaging in mice has shown that dendritic spines in contact with microglia are significantly more than in controls, which were untouched by microglia [41]. Microglia can also regulate the formation of synapses by secreting cytokines [42]. The above evidence suggests that microglia remove weaker synapses through synaptic pruning during synapse formation, which helps neurons mature, enhances synaptic signal transmission, and ensures normal brain function. The classical complement cascade pathway is the main pathway for microglia to exert synaptic pruning. This pathway is an innate immune surveillance mechanism that mainly functions through two components, C1q and C3. The pathway starts a cascade reaction through C1q, and finally C3 wraps the components that need to be cleared and attracts immune cells with C3 receptors to play an immune clearance role. Microglia are the only cell type among the resident cells in the CNS that carry C3 receptors [43]. In addition to the complement cascade, microglia also prune unnecessary synapses by binding the S4 isoform of GPR56 to everted phosphatidylserine (PS) on synapses [44]. The gene encoding the phosphatase and TENsin homolog (PTEN) is a risk gene for ASD, and ASD patients with PTEN mutations often exhibit macrocephaly. In the Ptenm3m4/m3m4 mouse model, PTEN is only expressed in the cytoplasm, leading to ASD-like features in mice. Studies have found that in this model, the expression of Iba1 and C1q is increased in Ptenm3m4/m3m4 microglia, enhancing their phagocytic ability, indicating that these microglia are in an activated state and the complement function is exerted. Coculturing neurons with Ptenm3m4/m3m4 microglia has revealed that these microglia have stronger synaptic pruning ability compared to the wild type [45]; this indicates that Pten mutations in ASD affect the complement cascade pathway of microglia, and the abnormal activation also leads to abnormal synaptic pruning function during neural development. Transmembrane protein 59 (TMEM59) can regulate microglial activity and neuroinflammation. Studies have shown that the expression of TMEM59 is decreased in autism patients. Knocking out TMEM59 in mice also resulted in ASD-like behavior, and defects in TMEM59 in microglia can disrupt the stability of the C1q receptor CD93, impairing its synaptic phagocytic ability, leading to enhanced excitatory neurotransmission and increased dendritic spine density [46]. An early transcriptome analysis showed a reduced expression of the C1q gene in brain neurons of RTT patients [47], suggesting that the complement cascade pathway is also abnormal in RTT. These findings suggest that ASD-related risk genes are related to the complement cascade pathway of microglia and mainly rely on C1q to function. The abnormal expression of this receptor can cause abnormal synaptic phagocytosis of microglia, leading to pathological phenotypes such as increased dendritic spine density. Abnormalities in the complement cascade pathway of microglia may be one of the reasons for impaired synapse formation and pruning processes in ASD.

#### 3.2.2. Neural Circuit Formation and Signal Pathway

Numerous studies have shown that patients with autism have local overconnections and cortical underconnections in their brains, which may be one of the causes of ASD [48]. Recent research reports that functional connections between social-related brain areas and many other brain areas are significantly weakened in ASD and are negatively correlated with social impairment [49]. During development, microglia play an essential role in forming neural circuits [50]. Studies have shown that under physiological conditions, “resting” microglia briefly detect synaptic status through contact with synapses. However, under pathological conditions, the contact time between microglia and synapses is prolonged, which may activate microglia to clear damaged synapses and increase synaptic turnover rate [39]. The ASD risk gene *SHANK3* encodes the multidomain scaffolding protein Shank3, which is highly expressed at postsynaptic sites and interacts with some synaptic scaffolding proteins to maintain the normal function of neurons and neural circuits. The deletion of Shank3 in the neocortex causes impaired neural circuits in multiple brain regions, leading to the simultaneous loss of synaptic and intrinsic homeostatic plasticity in pyramidal neurons of the neocortex. Research indicates that lithium has a certain effect in treating many neurological disorders, and that the effects of Shank3 mutations can be rescued by in vitro lithium treatment or by pharmacological inhibition of the lithium target glycogen synthase kinase 3 (GSK3) [51]. The upregulation of the RNA expression of another ASD risk gene, *CNTNAP2*, coincides with the onset of cortical synapse formation and circuit formation. Its encoded CASPR2 protein has been shown to stabilize formed synapses, and the lack of CASPR2 can cause dendritic spine abnormalities, synaptic dysfunction, excitatory–inhibitory imbalance, and other abnormalities. Mouse models with Cntnap2 mutations exhibit typical ASD phenotypes. In the Cntnap2 knockout mouse model, it was found that microglial activation could disrupt the development of neural circuits related to social behavior through an excessive pruning of synapses at critical developmental periods [52]. The use of oxytocin can strongly stimulate many brain regions, normalize connectivity patterns, and rescue social deficits in Cntnap2 knockout mice [53]. RTT is a neurodevelopmental disorder caused by mutations in the X-linked gene *MECP2*. In RTT, mutated MeCP2 in microglia leads to excessive glutamate secretion, causing neurotoxicity and resulting in neuronal damage or even death [54]. A study on the homeostasis of RTT neural circuits by microglia found no obvious synaptic abnormalities in the early onset (40 days) of RTT mice. However, in the later stages of the disease (56 days), microglia in RTT mice showed obvious excessive phagocytosis, leading to abnormalities in the neural circuits of RTT mice [55]. These works demonstrate that neural circuit connections are affected in syndromic ASD. Excessive microglial activation and phagocytic pruning of synapses lead to abnormal neural circuits. This effect can be reduced by using relevant drugs. The overactivation of the phosphatidylinositol-3-kinase (PI3K)/AKT signaling pathway during brain development plays a vital role in autism pathogenesis. The PI3K/AKT signaling pathway is mainly involved in neural activities such as synaptic plasticity, neuronal development, memory consolidation, and protein synthesis [56]. The overexpression of the PI3K/AKT signaling pathway increases cell death, neuroinflammation, and oxidative stress leading to neurological diseases [57]. Research has shown that chrysophanol (CPH) can improve autismlike symptoms by inhibiting the PI3K/AKT/mTOR pathway [58]. Additionally, studies have indicated that progranulin (PGRN) reduces autismlike symptoms by activating the PI3K/Akt/GSK-3β pathway [59], suggesting that in ASD, the activation of the PI3K/Akt pathway may not be entirely harmful, and different downstream targets of the PI3K/Akt pathway may have different effects on ASD. Apart from influencing neurogenesis and circuit construction in the CNS, microglia can maintain neuron survival by activating protein kinases, PI3K/Akt pathways, and Notch signaling pathways [30,60]. However, researchers have yet to study the PI3K/AKT signaling pathway of microglia in ASD. This pathway may be one of the critical signaling pathways for synapse formation and pruning by microglia in ASD.

## 4. Factors Released by Microglia in ASD

Microglia can promote brain development and neural repair by secreting a variety of factors to regulate neuronal activity and synaptic pruning [61] (Table 1). For example, tumor necrosis factor-α (TNF-α) promotes the proliferation and differentiation of neural progenitor cells [62], while brain-derived neurotrophic factor (BDNF) and insulin-like growth factor 1 (IGF-1) can enhance the development and maintenance of neural circuits [63]. Changes in the expression of these proteins may contribute to ASD. In the following sections, we focus on the impact of abnormal proteins’ expression related to microglia on ASD, aiming to provide insights for future research.

### 4.1. Deficient BDNF Associated with Microglia Causes Abnormal ASD

Brain-derived neurotrophic factor (BDNF) is found in almost all areas of the brain [76] and is primarily secreted by neurons, although it has also been detected in glial cells, including microglia [77]. There are two main isomers of BDNF: proBDNF, which mainly promotes neuronal apoptosis, clearing damaged and dysfunctional neurons, and synaptic pruning in the nervous system; while mBDNF mainly plays a role in supporting nerve and gliogenesis and developmental functions [78]. Studies have shown that under physiological conditions, microglia can produce and secrete both proBDNF and mBDNF, which bind to different receptors on the surface of neurons and affect synaptic plasticity [40]. However, under pathological conditions, microglia-derived BDNF contributes to synaptic remodeling and neuronal activity induced by nerve injury [79], indicating that BDNF secreted by microglia mainly regulates synaptic functions. BDNF can promote the formation and maturation of glutamatergic and GABAergic synapses during postnatal development. Valproic acid (VPA) is an antiepileptic drug, and its intake during pregnancy can affect the neurodevelopment of offspring, significantly increasing the risk of ASD in the offspring. Offspring exposed to VPA during pregnancy in rodent models exhibit typical ASD symptoms, including social deficits and repetitive behaviors. In ASD mouse models treated with VPA, an excitatory–inhibitory imbalance was found, and the expression of BDNF was severely reduced. Exogenous BDNF treatment during the critical period of excitatory–inhibitory imbalance can rescue synaptic function and autismlike behavior, indicating that synaptic function and autistic behaviors in ASD may be due to excitatory–inhibitory imbalance caused by BDNF deficiency during development [74]. One study demonstrated that BDNF reactivity is increased in ASD patients, and this increase is due to increased proBDNF, but proBDNF levels are reduced in ASD patients who receive mood stabilizing drugs. This suggests that autistic patients normalize BDNF levels with mood-stabilizing drugs, in which proBDNF plays a key role58. Some studies have measured serum levels of chemokine ligand 2 (CCL2) and chemokine ligand 5 (CCL5), as well as BDNF levels in children with ASD, and found that their cognitive function and behavioral deficits were associated with elevated CCL5 levels, while BDNF levels were significantly reduced. This shows that changes in immune function and a lack of neurotrophic factors are related to the pathogenesis of ASD [75]. Exposing pregnant rodents to lipopolysaccharide (LPS) during the critical period of pre- and postnatal (PN) neurodevelopment may cause changes in neurodevelopment, leading to neurodevelopmental disorders and resulting in an autism-like phenotype in rodents. Lower levels of BDNF were also observed in the brain regions of this ASD mouse model. This was more obvious in young mice, and only LPS-stimulated females showed obvious expression of the microglial activation marker Iba-1 in the hippocampus [80]. Reprogramming the arginase (Arg)-1+ microglia phenotype in the dentate gyrus of the hippocampus in ASD can increase the expression of BDNF, restore hippocampal neurogenesis, and improve depressive-like behavior [81]. According to existing studies, insufficient BDNF secreted by microglia can lead to impaired neurogenesis in ASD, and restoring BDNF expression can rescue part of the phenotype.

### 4.2. Reduced IGF-1 Associated with Microglia Causes Abnormal ASD

During the development of the nervous system, insulin-like growth factor-1 (IGF-1) plays a crucial role in promoting various aspects such as neural precursor cell differentiation [82], neuronal proliferation [83], synaptogenesis development [84], and neuronal survival [85]. IGF-1 is produced locally by neurons and glial cells in the brain, with microglia being the main source [86,87]. It acts through the IGF-1 receptor (IGF-1R) on neurons and astrocytes [88], and CD11+ microglia have been shown to regulate myelination in the CNS by secreting IGF-1 [89]. Due to its ability to cross the BBB [90], IGF-1 has gained attention as a potential therapeutic for neurological diseases, including ASD [91]. Studies have found significant associations between promoter polymorphism, the serum concentration of IGF-1, and ASD [92]. While serum IGF-I levels gradually increase with age in prepubertal and early adolescent children, the expression of IGF-1 is reduced in ASD. The disruption of IGF-1 signaling, including MAPK and PI3K/AKT1 signaling pathways, is considered one of the pathogenic mechanisms of ASD [73]. The PI3K/Akt pathway has been discussed above. The MAPK signaling pathway plays an important role in the normal development of the central nervous system and brain maturation. Research indicates that this pathway is associated with many genetic and psychiatric disorders. Among the MAPK signaling pathways, the ERK/MAPK pathway is the most widely studied. Increasing evidence suggests that there is a widespread dysregulation of the ERK/MAPK pathway in ASD. Furthermore, ASD-related risk genes such as neurofibromatosis type 1 (NF1), SynGAP, and FXS are all associated with elevated ERK/MAPK signaling, indicating that the dysregulation of this pathway may play an important role in the pathogenesis of ASD [93]. Treatment with IGF-1 has shown improvements in ASD symptoms, such as social withdrawal, repetitive behaviors, hyperactivity, and sensory reactivity, particularly in syndromic ASD like Phelan–McDermid syndrome [94]. Hyperbaric oxygen therapy (HBOT) has been reported to improve neuroinflammation and social behavior in ASD by reducing the number of Iba1-positive cells, upregulating the expression of IGF-1, and inducing neuroplasticity effects [95]. These findings suggest that ASD is associated with increased microglia numbers, and HBOT can reduce microglia numbers while increasing IGF-1 expression in microglia. This may lead to tissue repair, reduced neuroinflammation, and improved behavioral phenotype in ASD.

### 4.3. Increased TNF-α Associated with Microglia Causes Abnormal ASD

Tumor necrosis factor-α (TNF-α) is an inflammatory factor that is secreted by microglia in response to brain inflammation [96]. High concentrations of TNF-α have been found to induce apoptosis of NSCs/NPCs, while low concentrations of TNF-α have the opposite effect by promoting NSC proliferation and differentiation [62]. This dose-dependent effect may be attributed to the binding of TNF-α to different receptor subtypes. Binding to TNF-α receptor 1 inhibits neurogenesis, while binding to TNF-α receptor 2 promotes neurogenesis [97]. Studies have shown that patients with autism have increased levels of TNF-α and reduced expression of tumor necrosis factor α-regulated long noncoding RNA (THRIL) in their serum. Furthermore, the blood concentration of TNF-α is positively correlated with the severity of ASD symptoms [72]. Emerging evidence suggests that the activation of microglia triggered by maternal immune response may play a key role in the development of ASD. Studies have demonstrated that repeated allergic asthma during pregnancy can alter the expression of cytokines in the fetal environment, including interferon-γ (IFN-γ), granulocyte-macrophage colony-stimulating factor (GM-CSF), interleukin 1α (IL-1α), interleukin IL-6, and tumor necrosis factor α (TNF-α). These changes lead to homeostatic and neuroinflammatory alterations in the fetal brain [64]. The aforementioned evidence indicates that abnormal microglial activation in ASD affects the secretion of TNF-α, disrupting brain homeostasis and leading to neuroinflammation. Therefore, TNF-α may play a crucial role as a cytokine in the pathological process of ASD.

## 5. The Immune Function of Microglia on ASD

### 5.1. Migration and Monitor of Microglia

There are two types of tissue macrophages in the CNS: microglia located in the brain parenchyma and macrophages located at the CNS barrier, including perivascular macrophages, meningeal macrophages, and choroid plexus macrophages. [98]. These cells play an essential role in maintaining nervous system homeostasis and preventing excessive immunity in various nervous system states. During early CNS development, microglia migrate into the brain and colonize various areas of the brain [99]. After migration is complete, microglia replenish themselves primarily through self-proliferation [100]. This migratory activity of microglia in immune responses is mediated by ATP and proceeds rapidly through interactions with blood vessels in response to abnormal situations in the CNS [12,101]. Mitochondria, known as “energy factories” in organisms, are the primary source of ATP. Research has confirmed a connection between mitochondria and immune responses. Mitochondria can affect immune responses, and immune responses can also affect the function of mitochondria [102]. Strong evidence suggests that oxidative stress may be the primary cause of ASD. In ASD patients, oxidative stress levels are elevated, and genes encoding key enzymes in the reactive oxygen species (ROS) scavenging system have significantly reduced expression. This reduction may contribute to ASD and lead to increased susceptibility to ROS-mediated damage and neuronal toxicity [103]. In cells, the electron transport chain (ETC) in mitochondria is the primary source of ROS [104]. Mitochondrial dysfunction has been widely confirmed in ASD, with 5–80% of children with autism experiencing mitochondrial dysfunction, many of whom exhibit novel mitochondrial disorders [105]. Whether endogenous or exogenous oxidative stress causes mitochondrial dysfunction, dysfunctional mitochondria produce more ROS, further increasing oxidative stress levels. This increase leads to a vicious cycle due to the two aspects [106]. Forkhead box protein P1 (FOXP1) is mainly expressed in the cerebral cortex, hippocampus, and striatum, and plays an important role in regulating neurogenesis and synapse formation. In humans, a haploinsufficiency of FOXP1 leads to the FOXP1 syndrome, a neurodevelopmental disorder with prominent autistic features. FOXP1 is currently one of the top-ranked ASD risk genes. In animal models, Foxp1+/− mice exhibit phenotypes similar to patients. Studies have confirmed that increased oxidative stress levels and insufficient energy supply caused by mitochondrial dysfunction in FOXP1+/− mice are the basis of cognitive and motor dysfunction caused by FOXP1 [107]. Additionally, significantly increased levels of mitochondrial DNA were detected in the serum of children with ASD. Mitochondrial DNA is the primary activator of neuroinflammation and can cause neuroinflammation by stimulating microglia to release IL-1β. Abnormal microglia in mouse offspring can be improved by the transplantation of normally functioning mitochondria in MIA [108]. Therefore, mitochondrial dysfunction may play an important role in microglial abnormalities in ASD. However, whether it affects the immune function of microglia still requires more evidence.

### 5.2. Different Status of Microglia

Under physiological conditions, microglia in the brain are quiescent, highly branched, and have many tiny processes. They express low levels of immune-related molecules [109]. However, microglia are not completely static and constantly monitor the surrounding neural environment [110]. When abnormalities are detected, microglia undergo rapid morphological and functional changes, known as microglial activation [111]. Activated microglia can be classified into two types: classical activation (M1) mode and alternative activation (M2) mode [112], similar to macrophage activation [113]. M1 microglia have a proinflammatory phenotype, releasing inflammatory factors and chemokines that contribute to inflammation, neurotoxicity, and neuronal death. On the other hand, M2 microglia release anti-inflammatory and neuroprotective factors, promoting tissue repair and maintenance [29,114,115]. In Alzheimer’s disease (AD), increased levels of ROS switch microglia from M2 to M1, exacerbating the disease’s pathological phenotype [116], this highlights the importance of regulating microglial activation patterns for normal brain function and the prevention of neurological diseases [117]. Simultaneously inhibiting M1 microglia and activating M2 microglia could be effective therapeutic targets [118]. Studies have shown that the induction of MIA in mice leads to an ASD-like phenotype in offspring, accompanied by a significant increase in microglia in fetal and neonatal neurogenic areas. Under normal circumstances, microglia consist of M1 (CD86+/CD206−) and mixed M1/M2 (CD86+/CD206+)-like subpopulations. However, MIA reduces the M1 phenotype and increases the M1/M2 mixed phenotype microglia. In the subventricular zone (SVZ) of neonatal mice, there is a significant increase in neural progenitor cells, accompanied by changes in the expression of multiple cytokines [35]. This suggests that in early ASD development, microglia tend to activate the M2 mode, leading to abnormal neurogenesis and potentially worsening the ASD phenotype. The transcriptome analysis of isolated microglia from RTT mice also indicates that a loss of MeCP2 affects the balance between M1 and M2 patterns [119]. However, it is not yet known which activation mode is dominant in RTT. These findings suggest that the imbalance of microglial M1 and M2 patterns in syndromic ASD can impact phagocytosis and neurogenesis. Restoring the balance of microglial activation patterns may be a potential method for treating ASD.

## 6. Interaction between Microglia and Other Cells in the CNS

### 6.1. Interaction between Microglia and Neurons

Microglia–neuron interactions have a crucial role in neural proliferation, differentiation, maturation, and circuit formation [120], as previously explained (Figure 2). The impact of these interactions on CNS diseases has been extensively studied [121,122,123]. Fractalkine-CX3CR1 signaling is a fundamental pathway regulating microglia–neuron interactions [124]. A study has confirmed that CX3CR1-deficient microglia cannot maintain the survival of cortical neurons [60]. Research on CX3CR1 knockout mice found that CX3CR1 knockout led to a decrease in microglia in the brains of early postnatal mice, suggesting that CX3CR1 is a key factor in the normal migration ability of microglia [125]. Fractalkine is mainly expressed by neurons, and functional CX3CR1 is expressed by microglia. Studies have found that both fractalkine and CX3CR1 are involved in the response of motor neurons to injury after peripheral nerve injury occurs [126]. Research has shown that in the LPS-induced ASD mouse model, a significant increase in dendritic spine density of granule cells in the hippocampus was observed, and the expression level of CX3CR1 in the hippocampus decreased. This suggests that the interaction between microglia and neurons is reduced in the hippocampus of the offspring of MIA mice, indicating that synaptic pruning mediated by microglia–neuron interactions in the brains of offspring mice may be affected [127]. Another study using CX3CR1 exon-targeted sequencing and genetic association in ASD patients found a statistically significant correlation between the Ala55Thr variant in CX3CR1 and the ASD phenotype. An in vitro functional analysis also showed that this variant inhibited fractalkine-CX3CR1 signaling and weakened microglia–neuron interactions. People with the Ala55Thr variant of CX3CR1 have an increased probability of developing ASD [128]. The above evidence suggests that neuron–microglia interactions are affected in ASD. A study on RTT found that after knocking out CX3CR1 expression in RTT mice, the morphology of neurons and microglia in the mouse brain was restored, the pathological phenotype of RTT was improved, and the expression of IGF-1 was increased, causing enhanced neurotrophic effects [129], indicating that neuron–microglia interactions in RTT aggravate the pathological phenotype of RTT. Further evidence is needed to determine whether the neuron–microglia interaction mediated by fractalkine-CX3CR1 plays a harmful or beneficial role in the pathological process of ASD.

### 6.2. Interaction between Microglia and Astrocyte

Astrocytes are highly branched cells with a complex morphology and are in extensive contact with other cells. They play a role in the CNS by regulating nervous system homeostasis, synapse formation/pruning/regulation, neurovascular coupling, and participating in the BBB and many other functions [130]. They mainly affect neurogenesis and development by releasing growth factors (BDNF, FGF-2, GDNF, VEGF, etc.), ATP, lactic acid, interleukins, etc. [131]. In terms of synaptic transmission of neurons, astrocytes form “tripartite synapses” by contacting presynaptic and postsynaptic neurons to enhance or weaken neural signal transmission [132]. In addition to neurons, astrocytes also provide nutritional factors for the growth and development of microglia and promote their homeostasis and function [133]. Furthermore, astrocytes coordinate with microglia in clearing apoptotic cells and cell debris. Astrocytes detect excess synapses and release IL-33 during synapse pruning, which recruits microglia for the subsequent phagocytosis of synapses [132]. Under physiological conditions, adjacent astrocytes can absorb and circulate neurotransmitters and metabolites through gap junctions, maintaining ionic homeostasis in the nervous system to control neuronal excitability and synaptic transmission. Microglia are essential for the proper functioning of such astrocyte networks [134]. Glial fibrillary acidic protein (GFAP) is a type III intermediate filament protein mainly expressed in astrocytes. Studies have reported that the concentration of GFAP in the cerebrospinal fluid of ASD patients is three times higher than that of controls [135]. At the gene expression level, GFAP shows a brain region specificity, while studies at the protein level found that the expression level of GFAP was upregulated in the prefrontal lobe, parietal lobe, cerebellum, and cingulate gyrus [136]. It should be noted that GFAP is not a marker of the number of astrocytes but a marker of reactive astrocytes. Studies have pointed out that in ASD, the number of astrocytes is reduced while activated astrocytes increase significantly [137]. When neuroinflammation occurs, astrocytes become reactive astrocytes after being damaged and establish an immune response through morphological changes and proliferation. This not only has harmful effects such as aggravating neuroinflammation and damaging synapses but also has beneficial effects such as neuroprotection [108]. In the inflammatory response, microglia and astrocytes have a bidirectional effect, and abnormalities in this interaction can also lead to neuronal dysfunction [138]. Studies have shown that LPS-activated microglia induce reactive astrocytes, inducing their neurotoxic phenotype. At the same time, microglia are activated by reactive astrocytes, and ATP released by microglia activated by astrocytes leads to a further exacerbation of inflammation. This interaction can also simultaneously polarize microglia and astrocytes into a neuroprotective phenotype, which can play a beneficial role [132]. Studies in ASD patients and animal models have shown that isolated cortical microglia and astrocytes exhibit a reactive morphology, increased proinflammatory cytokines, and impaired miRNA processing. In microglia–astrocyte coculture, microglia from ASD-model animals exhibit reactivity and exacerbate astrocyte reactivity [139]. The valproic acid (VPA)-induced ASD mouse model also confirmed that the interaction between microglia and astrocytes was closely related to neuroinflammation in ASD. Astrocytes isolated from the VPA animal model exhibit a persistent proinflammatory phenotype, and microglia exacerbate astrocyte reactivity. This interaction does not allow microglia to engage in response to phagocytic stimuli, promoting neuroinflammation in ASD [139]. Abnormal astrocyte Ca^2+^ signaling in ASD has been identified as the mechanism responsible for ASD-specific phenotypes and neuronal defects [140]. IP3R2 knockout causes ASD-like behaviors in mice, confirming the important role of abnormal astrocyte calcium signaling in ASD [141]. Astrocytes regulate ATP expression and release in a calcium-dependent manner, and astrocyte-derived ATP is implicated in a potential mechanism underlying the ASD phenotype in IP3R2 knockout mice. Thus, the P2X2 receptor has been identified as an important mediator in this mechanism [141]. Perhaps in the future, it may be possible to improve the harmful interactions between astrocytes and microglia by affecting this receptor, thereby improving the pathological process of ASD.

### 6.3. Interaction between Microglia and Oligodendrocyte

Besides astrocytes and microglia, oligodendrocytes are another major glial cell type in the CNS. Oligodendrocytes form myelin, which is a multilayered lipid membrane structure that wraps around axons, increasing the propagation speed of action potentials and providing metabolic and nutritional support to axons [142]. Research shows that microglia can promote the survival and differentiation of oligodendrocytes [143,144]. In vitro, M1 pattern microglia produce cytotoxicity to oligodendrocytes, while M2 pattern microglia exert a phagocytic activity. Such phagocytic activity can promote oligodendrocyte differentiation to support the remyelination of the CNS [143,145]. The “microglial fountain” describes the spatiotemporal characteristics of a group of microglia that appear in the periventricular region during the early postpartum period. This group of microglia affects oligodendrocyte cells (OPCs) and is important for remyelination [144]. This group of microglia also impacts oligodendrocytes in the early neonatal brain by secreting IGF-1 and other cytokines related to myelination [146]. So far, the mechanism of microglia–oligodendrocyte interaction has not been explored, but it is certain that the influence of microglia on OPCs in the early postnatal period ensures the normal development of myelin. Microglia also have an important influence on oligodendrocytes. A magnetic resonance imaging (MRI) study of 21 children with ASD (1.5–5.5 years old) during critical neurodevelopmental periods using T1-weighted (T1W) and T2-weighted (T2W) ratios to assess myelin in vivo showed that the control group exhibited age-related increases in myelin content during early childhood and preschool. Such a phenotype was not observed in those with ASD, suggesting that myelination in the brains of children with autism is affected early in life [147]. In addition, studies on VPA-induced ASD mouse models found hypomyelination in brain areas related to social behavior (amygdala and piriform cortex), and a decrease in the number of oligodendrocyte lineage cells and mature oligodendrocytes was also observed in the piriform cortex [148]. These studies confirm that oligodendrocytes and myelination in the brain are affected during ASD pathology. The study found that after LPS stimulation, the ASD-related genes chromatin domain helicase DNA-binding protein 8 (CHD8) and tuberous sclerosis complex 2 (TSC2) were silenced, and the expression of IGF-1 mRNA was changed in microglia. TSC2 deficiency inhibits the phagocytic activity of microglia. In vitro coculture experiments with TSC2 deficiency NSCs and microglia, and CHD8 and TSC2 knockout mouse models both found that microglia-mediated oligodendrocyte development was inhibited [149]. The above evidence shows that in ASD, microglia affect oligodendrocyte differentiation and development, which may further affect myelination in the ASD brain. More evidence remains to be further elucidated.

## 7. Conclusions

There is growing evidence that microglia are important in neurogenesis, circuit formation, and immune function. During the neurodevelopmental process of ASD, abnormal neurogenesis and synapse formation can lead to neural circuit disorders, resulting in ASD-like behavioral manifestations. At present, more and more studies have found that this process involves neuroinflammation and microglia activation. Overactivated microglia can be affected by ASD-related risk genes and have abnormal functions, such as damaged synaptic pruning function. Factors secreted by microglia also play a key role in the neurodevelopment process. For example, changes in the expression of BDNF, IGF-1, and TNF-α can lead to excessive neurogenesis, abnormal synaptogenesis and development, and aggravated ASD-like behaviors. As immune cells, the ability of microglia to migrate and recognize phagocytic signals cannot be ignored. The supply of mitochondrial energy plays an important role in the migration process of microglia, and microglia migration is inseparable from the immune response. The activation of microglia with different morphologies also affects neurogenesis and development. However, the underlying causes of neurodevelopmental abnormalities and immune dysfunction in ASD remain elusive. In summary, we started with aspects such as the origin of microglia, neurogenesis and development, and cell-to-cell interactions, and related the discussion of microglia to some pathological processes in the development of ASD, proposing some possible mechanisms of microglia in the pathogenesis of ASD. In the past, the biggest limitation in research on microglia was the difference between human microglia and those of other species [150]. This difference may have caused research on microglial diseases in model animals to be unable to truly simulate the symptoms of patients. In addition, a transcriptome analysis shows that microglia derived from human induced pluripotent stem cells can only exhibit a phenotype similar to microglia, rather than true microglia [150]. The emergence of organoid culture technology has made up for this limitation [151]. The use of human brain organoids to conduct microglial research on ASD may help us understand microglia in ASD to a great extent, thereby further exploring the pathogenesis of ASD.

## Figures and Tables

**Figure 1 biomedicines-12-00210-f001:**
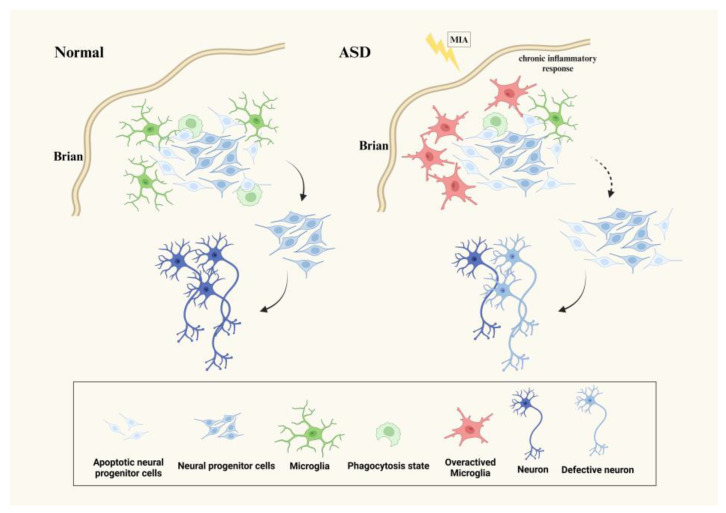
Microglia directly regulate neurogenesis through phagocytosing neural stem cells during neurogenesis, and only the last surviving neural progenitors can continue to differentiate into mature neurons. MIA causes ASD-like behavior in offspring, triggers a chronic proinflammatory response during fetal development. The overactivation of microglia can negatively regulate synaptic pruning and neurogenesis by altering the expression of key phagocytic genes. In ASD, the abnormal activation of microglia can result in abnormal nerve development and impaired neuronal function.

**Figure 2 biomedicines-12-00210-f002:**
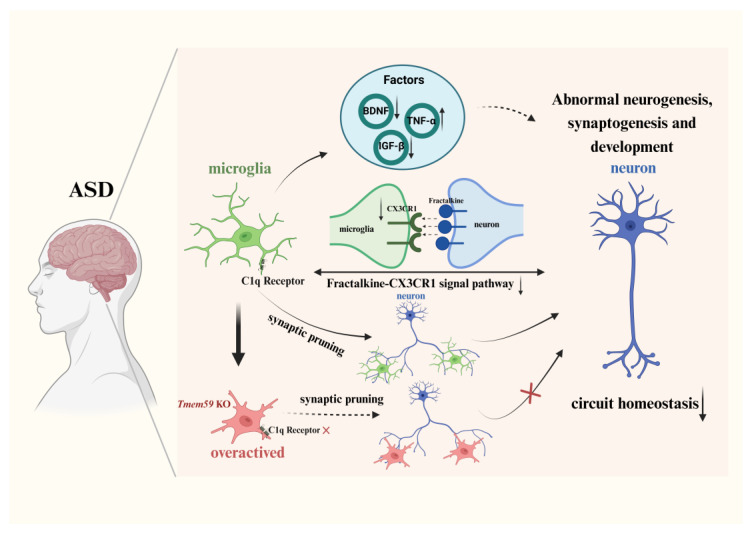
Microglia play a crucial role in synaptic pruning and the maintenance of circuit homeostasis in ASD. They promote brain development and nerve repair, regulate neuronal activity, and prune synapses by secreting a variety of factors. In ASD brains, microglia secrete abnormal factors, such as reduced levels of BDNF, which leads to impaired synaptic function. The reduced expression of IGF-1 and increased secretion of TNF-α increase neuroinflammation. Alterations in growth factors can alter homeostasis in the brain and induce ASD-like behaviors. Due to the reduced expression of CX3CR1 and inhibition of fractalkine-CX3CR1 signaling, microglia–neuron interactions are reduced in ASD. This reduction also affects synaptic pruning mediated by microglia–neuron interactions, causing impaired neuronal function. In addition, microglia can also prune synapses through the complement cascade pathway and mainly rely on C1q and C3 receptors to function. Tmem59 KO mice, which induce ASD-like behavior, have abnormal expression of C1q receptor and lead to abnormal synaptic phagocytosis of microglia, resulting in abnormal synaptic pruning function during neural development.

**Table 1 biomedicines-12-00210-t001:** Changes in the expression levels of factors secreted by microglia and their effects on ASD.

Factor	Species	Area and Expression Level	Influence	Reference
IL-1α	Offspring of MAA mice	Brain of offspring ↑	Passed through the placenta, induces ASD in offspring	[64]
IL-1β	ASD rat	Blood, brain ↑	Hippocampal and cortical neuronal death, impairment of axonal integrity and myelination	[65]
	ASD patient	ACC ↑	Neurons decrease in size and number, and synaptic connections are damaged	[66]
	ASD rat	Cerebral cortex ↑	Accompanied by microglial state transition inducing the ASD-like phenotype	[67]
IL-6	ASD patient	Plasma ↑	Promote oxidative stress and neuroinflammation, leading to dysfunction in ASD	[68]
	Offspring of MAA mice	Brain of offspring ↑	Passed through the placenta, induces ASD in offspring	[64]
IL-37	ASD patient	Amygdala, prefrontal cortex ↑	Inhibits the secretion of proinflammatory cytokines IL-1β and CXCL8 by cultured microglia in vitro	[69]
IL-38	ASD patient	Amygdala ↓	Inhibits the secretion of proinflammatory factors from cultured microglia in vitro	[70]
TNF-α	ASD Rat	Blood, brain ↑	Hippocampal and cortical neuronal death, impairment of axonal integrity and myelination	[65]
	Offspring of MAA mice	Brain of offspring ↑	Passed through the placenta, induces ASD in offspring	[64]
	ASD Rat	Cerebral cortex ↑	Accompanied by microglial state transition inducing the ASD-like phenotype	[67]
	HI Offspring of MIA mice	Brain ↑	Impairing prominent network connections in offspring mice; induces ASD	[71]
	ASD patient	Serum ↑	Expression levels are positively correlated with ASD severity	[72]
IFN-γ	Offspring of MAA mice	Brain of offspring ↑	Passed through the placenta; induces ASD in offspring	[64]
GM-CSF	Offspring of MAA mice	Brain of offspring ↑	Passed through the placenta; induces ASD in offspring	[64]
IGF-1	ASD patient	Anterior cingulate cortex↓	Neurons decrease in size and number, and synaptic connections are damaged	[66]
	ASD patient	Serum ↓	Affects synaptic connections by activating MAPK and PI3K/AKt signaling pathways	[73]
NF-κB	HI offspring of MIA mice	Brain ↑	Impairing prominent network connections in offspring mice induces ASD	[71]
BDNF	ASD mice	ACC ↓	Excitatory/inhibitory imbalance, impaired synaptic plasticity and function	[74]
	ASD patient	Serum ↓	Expression levels are negatively correlated with ASD severity	[75]
	ASD patient	Serum ↓	Affects synaptic connections by activating MAPK and PI3K/Akt signaling pathways	[73]

Abbreviations: interleukin (IL); maternal asthma allergy (MAA); anterior cingulate cortex (ACC); tumor necrosis factor-α (TNF-α); interferon-γ (IFN-γ); maternal immune activation (MIA); hypoxia ischemia (HI); granulocyte-macrophage colony-stimulating factor (GM-CSF); insulin-like growth factor-1 (IGF-1); nuclear factor kappa-B (NF-κB); brain-derived nutritional factors (BDNF). ↑: increase; ↓: decrease.

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
