# Peer review of "The Impact of Microglia on Neurodevelopment and Brain Function in Autism"

_biomedicines, 2024, doi:10.3390/biomedicines12010210_

Round 1

Reviewer 1 Report

Comments and Suggestions for Authors

This article summarizes recent research on the effects of microglia on neurodevelopment in ASD. It is particularly noteworthy for its focus on the link between microglial function and neurodevelopmental abnormalities and will be very useful for a research field on glial cell-related signaling pathways in ASD.

Minor points I have just realized are as follows.

Line 144-145, "In vivo... ...more than in controls". I could not see which cases were compared to the control.

Line 350, MIA 110 -> MIA [110]

Line 466, ...a beneficial role 140. ->  ...a beneficial role [140]

Line 477, P3R2 -> IP3R2

Author Response

Dear Reviewers,

Thank you very much for your time involved in reviewing the manuscript and your very encouraging comments on the merits, giving us the opportunity to submit a revised draft of the manuscript “Reminder to Autism Spectrum Disorder: Development and function of microglia” for publication in the Journal of Biomedicines (ISSN: 2227-9059). We have incorporated most of the suggestions made by you. Those changes are highlighted in the manuscript. Please find the detailed responses below and the corresponding revisions in the re-submitted files. All page numbers refer to the revised manuscript file with tracked changes.

Comments:

“This article summarizes recent research on the effects of microglia on neurodevelopment in ASD. It is particularly noteworthy for its focus on the link between microglial function and neurodevelopmental abnormalities and will be very useful for a research field on glial cell-related signaling pathways in ASD.”

We also appreciate your clear and detailed feedback and hope that the explanation has fully addressed all of your concerns. In the remainder of this letter, we discuss each of your comments individually along with our corresponding responses.

Comments 1: Line 144-145, "In vivo... ...more than in controls". I could not see which cases were compared to the control.

Response 1: Thank you for pointing this out. We agree with this comment. We really didn't explain the control group clearly. Microglia in contact with dendrites often form new dendritic spines. In this case, the control group is those dendritic branches that the microglia are not close to. Therefore, we have revised that Line L159-161, "In vivo... ...more than in controls". -> “In vivo imaging in mice showed that dendrites in contact with microglia would generate significantly more dendritic spines than dendrites not in contact with microglia”.

Comments 2: Line 350, MIA 110 -> MIA [110]

Response 2: Agree. We have revised the format to emphasize this point.

Comments 3: Line 466, ...a beneficial role 140. ->  ...a beneficial role [140]

Response 3: Agree. We have revised the format to emphasize this point.

Comments 4: Line 477, P3R2 -> IP3R2

Response 4: Agree. We have revised the format to emphasize this point.

We would like to take this opportunity to thank you for all your time involved and this great opportunity for us to improve the manuscript. We hope you will find this revised version satisfactory.

Yours sincerely,

Zhengbo Wang

State Key Laboratory of Primate Biomedical Research; Institute of Primate Translational Medicine, Kunming University of Science and Technology, 650500, Kunming, China.

Reviewer 2 Report

Comments and Suggestions for Authors

In this review paper, the authors described the contribution of microglia to autism spectrum disorder. They paid attention on their role in abnormal brain development, neurogenesis, synaptic pruning, formation of neuronal circuits and intracellular signalling, neuroinflammation and cytokines release, levels of BDNF-1 and IGF-1, and interactions with other types of glial cells.

1. Although the information provided could be considered as relevant, there are several other review papers covering the same topic. Some of them are:

https://www.ncbi.nlm.nih.gov/pmc/articles/PMC9714329/

https://www.frontiersin.org/articles/10.3389/fncel.2016.00021/full

https://www.mdpi.com/1422-0067/24/24/17297

https://api-journal.accscience.com/journal/article/preview?doi=10.36922/an.v1i3.167

https://karger.com/dne/article/37/3/195/108002/Role-of-Microglia-in-Autism-Recent-Advances

https://link.springer.com/article/10.1007/s00401-023-02629-2

The authors should provide explanation on the level of novelty in their manuscript.

2. L39: Large-scale genetic studies have revealed hundreds of risk genes associated with ASD, most of which are involved in brain development, neuronal activity, signal transduction, transcriptional regulation, and synaptic function [5]. Almost the same is unnecessarily repeated in L81 Studies have shown that ASD, as a developmental disorder of the nervous system, is associated with numerous risk genes related to brain development, neuronal activity, synaptic function, and more [5].

Based on the above paragraph, the authors should better explain the genetic background of the ASD and avoid repetitiveness.

3. Several animal models of ASD are mentioned in the text without any explanation (genetic background). Different mouse models of ASD should be briefly explained at their first appearance.

4. Figures in the text need higher resolution (like in Supplementary files).

5. PI3K/Akt pathway is not explained satisfactory.

L207: Overexpression of the PI3K/AKT signaling pathway increases cell death, neuroinflammation, and oxidative stress leading to neurological diseases [52]. Studies have shown that chrysophanol (CPH) can improve autism-like symptoms through the PI3K/AKT/mTOR pathway [53]. Additionally, progranulin (PGRN) can reduce autism-like symptoms through the PI3K/Akt/GSK-3ß pathway [54]. -the reference cited (54) shows neuroprotective effects of Akt activation

6. At least MAPK pathways should also be commented.

7. Throughout the manuscript many genes are mentioned, such as CShanks3,Cntnap2, ....the role of these genes and their potential involvement in ASD should be briefly introduced for better clarity

8. Some parts of the manuscript are more appropriate for the Introduction paragraph, e.g. L320-326, L354-364

9. L320 -As resident immune cells in the CNS, microglia not only maintain blood-brain barrier integrity – the role of microglia in the preservation of the BBB integrity should be explained more comprehensively

10. L420 – BDNF is not a cytokine, also in L528-530, this part of the manuscript should be written correctly

11. L464, L479 – it is unclear what types of cells release ATP, it should be written more clearly

There are also some other minor issues that should be additionally explained/improved.

The title of the manuscript is not well worded

Differences in terms of neural stem cells (NSCs), neural precursor cells and neural progenitor cells should be explained and used carefully.  For example, in L92 the authors mentioned NSCs, but neural progenitor cells are in the title of the cited paper, see also L293

The authors should pay attention throughout the manuscript that abbreviations are defined at the first appearance and then used thereafter in the abbreviated form, at several places in the manuscript the authors defined abbreviations again and again or forget to use abbreviations

L144 -In vivo imaging in mice has shown that dendritic spines in contact with microglia are significantly more than in controls [36]. – unclear

L168 - An early transcriptome analysis of female RTT patients showed that C1q gene expression in brain neurons of RTT patients was reduced [42], indicating that C1q in the complement cascade pathway is also abnormal in RTT patients. - repetition

L187 - Deleting the ASD risk gene Shank3 isoform in the neocortex leads to a simultaneous loss of neocortical pyramidal neuron synapses and intrinsic homeostatic plasticity, but extracorporeal lithium (Li) treatment or pharmacological inhibition of the Li target glycogen can rescue this [46]. – unclear why Li is mentioned, should be explained

L299, L309 – repetition

L315 - what is meant by CNS-associated macrophages found in the periphery, should be written more clearly

L329 -As we all know – I suggest deleting the phrase

There are also several typos, references cited without brackets, ….

Comments on the Quality of English Language

Minor to moderate English revision needed.

Author Response

Please see the Word.

Round 2

Reviewer 2 Report

Comments and Suggestions for Authors

Congratulations!

Comments on the Quality of English Language

Minor editing